


# Limitations of rainfall thresholds for debris-flow prediction in an Alpine catchment

Jacob Hirschberg[1,2], Alexandre Badoux[1], Brian W. McArdell[1], Elena Leonarduzzi[2,1], and Peter Molnar[2]

[1]Mountain Hydrology and Mass Movements, Swiss Federal Institute for Forest, Snow and Landscape Research WSL, Birmensdorf, Switzerland
[2]Institute of Environmental Engineering, ETH Zurich, Zurich, Switzerland

**Correspondence:** J. Hirschberg (jacob.hirschberg@wsl.ch)

**Abstract.** The prediction of debris flows is relevant because this type of natural hazard can pose a threat to humans and infrastructure. Debris-flow (and landslide) early warning systems often rely on rainfall intensity-duration (ID) thresholds. Unfortunately, no standardized procedures exist for the determination of such ID thresholds, and a validation and uncertainty assessment is often missing in their formulation. As a consequence, updating, interpreting, generalizing and comparing rainfall

thresholds is challenging. Using a 17-year record of rainfall and 67 debris flows in a Swiss Alpine catchment (Illgraben), we determined ID thresholds and associated uncertainties as a function of record length. Furthermore, we compared two methods for rainfall definition which consider both triggering and non-triggering events, based on linear regression and/or True Skill Statistic maximization. The main difference between these approaches and the well-known frequentist method is that non-triggering rainfall events were also considered here for obtaining ID-threshold parameters. Depending on the method

applied, the ID-threshold parameters and their uncertainties differed significantly. We found that 25 debris flows are sufficient to constrain uncertainties in ID-threshold parameters to ±30% for our study site. We further demonstrated the change in predictive performance of the two methods if a regional landslide data set was used instead of a local one, with important implications for ID-threshold determination. Furthermore, we tested if the ID-threshold performance can be increased by considering other rainfall properties (e.g. antecedent rainfall, maximum intensity) in a multivariate statistical learning algorithm based on decision

trees (random forest). The highest predictive power was reached when the 30-min maximum accumulated rainfall was added to the ID variables, while no improvement was achieved by considering antecedent rainfall for debris-flow predictions in Illgraben. Although the increase in predictive performance with the random forest model was small, such a framework could be valuable for future studies if more predictors are available from measured or modelled data.



## 1 Introduction

Debris flows are a common geomorphic process and hazardous phenomenon in mountain regions. They move rapidly as a surging flow of saturated debris. In contrast to other mass movements, such as shallow landslides, debris flows follow an established flow path, along which they often entrain substantial amounts of sediment and water stored in the channel (Hungr et al., 2014; de Haas et al., 2020). In Switzerland, where this study is conducted, landslides and debris flows caused 74 fatal accidents between 1946 and 2015 (Badoux et al., 2016). Globally, debris flows cause about 165 fatalities per year on average, with most of them occurring in mountainous regions of developing countries (Dowling and Santi, 2014). Furthermore, debris flows have the potential to damage property, infrastructure, managed forests and agricultural land (Hilker et al., 2009). Therefore, the development of early warning systems (EWS) for debris flows and other rapid gravitational mass movements, involving novel measurement techniques and models, is a priority in many countries (Stähli et al., 2015). EWS often rely on rainfall thresholds (Guzzetti et al., 2020). The most common rainfall thresholds are drawn in the rainfall duration ($D$) and mean rainfall intensity ($I$, or cumulative rainfall) space, taking the form $I = \alpha D^{-\beta}$, which is a linear curve in logarithmic space (Caine, 1980).

In alpine settings, debris flows mostly develop following a shallow hillslope landslide caused by increased pore water pressure (e.g. Iverson, 1997). Another cause can be runoff, where sediment deposits in the channel are mobilized as a mass movement (Takahashi, 1978, 1981) or sediments are progressively bulked up (Fryxwell and Horberg, 1943; Johnson and Rodine, 1984; Tognacca, 1999; Gregoretti, 2000). Physically-based models considering these mechanisms can be used to infer ID thresholds leading to debris-flow initiation (e.g. Berti and Simoni, 2005; Berti et al., 2020; Tang et al., 2019). However, because such models require a great deal of input data of high quality, empirically determined ID thresholds are still more common and are determined at the local, regional or global scale (e.g. Caine, 1980; Guzzetti et al., 2007; Coe et al., 2008; Badoux et al., 2009; Staley et al., 2013; Abancó et al., 2016; Bel et al., 2017).

The problem with the use of ID thresholds is that there is no standardized procedure for their determination, and they are rarely validated. Consequently, generalizing and updating ID thresholds is challenging, as is comparing between them to hypothesize about the possible site-related differences in geomorphology, lithology, terrain and soil properties that lead to a different response to rainfall forcing (Segoni et al., 2018). The sensitivities of ID thresholds have to be better understood and studied for such comparisons to be meaningful. Major uncertainties arise from various issues related to the quality of the rainfall record used. ID thresholds often rely on rainfall data from rain gauges, which can be located on the valley floor or in a neighbouring valley rather than in the immediate vicinity of the debris-flow initiation area. Studies in the Italian Alps have shown that in orographically complex areas, especially for short convective rainstorms, precipitation intensities can decay significantly (30–60%) within short distances (5–10 km) from the centre of the rainfall cell (Marra et al., 2016). This may lead to underestimations of $\alpha$ by up to 70%, and is one reason for the high false alarm rate of ID thresholds (Nikolopoulos et al., 2014). Another factor causing ID-threshold underestimation is the coarse temporal resolution of the rainfall data. Landslide and debris-flow data sets going far back in time, or relying on satellite-based estimates of rainfall, can be strongly affected by a coarse temporal resolution. Especially if events are triggered by short-duration storms (minutes to hours), the event mean





rainfall intensity is considerably underestimated when daily rainfall records are used. In a study with synthetic data, Marra
(2018) showed that using daily data results in significant threshold underestimation. Gariano et al. (2020) confirmed this effect
for a real case study in Italy. However, the accuracy of ID thresholds based on rainfall data with a sub-daily resolution can
also be limited, for example if the exact timing of the debris flows or landslides is unknown (Leonarduzzi and Molnar, 2020).
This is often the case, unless the area is closely monitored or damage was caused and immediately recognized. Therefore, in
studies where the landslide timing is only imprecisely known and sub-daily rainfall data are used, the entire rainfall event or
the rainfall until the highest intensity was reached is considered to be the triggering rainfall. This uncertainty in event timing
has been shown to lead to inflated triggering rainfall amounts and subsequently to overestimated ID thresholds (Staley et al.,
2013; Leonarduzzi and Molnar, 2020; Bel et al., 2017). Additional uncertainties stem from the discretization of the rainfall time
series into rainfall events. Rainfall events are usually separated by a minimum inter-event time (MIT), a period that is intended
to mark separate, independent rainfall periods. For studies in debris-flow torrents, the MIT can range from 10 min to 6 h and is
often chosen subjectively without a sensitivity analysis. Bel et al. (2017) combined different MIT values with uncertainty in the
timing of debris-flow detection in a French torrent. The obtained uncertainty bounds in ID thresholds encompassed almost all
ID thresholds previously published from other torrents. Despite their importance, uncertainties in ID curves are seldom used
in prediction.

The inaccuracies in rainfall data used to establish ID thresholds are one source of uncertainty leading to the high false alarm
rate. ID thresholds are also criticized for ignoring other information contained in the rainfall time series, such as peak intensities
and antecedent rainfall. For debris flows, peak intensities at high temporal resolution ($\leq$10 min) have been shown to have an
especially high predictive power (e.g. Abancó et al., 2016; Bel et al., 2017). Multivariate statistical methods, such as logistic
regression, have been tested and applied to improve the prediction of post-wildfire debris flows in the US (Cannon et al., 2010;
Staley et al., 2017), and for a French Alpine torrent (Bel et al., 2017). More advanced machine learning techniques are also
becoming an attractive tool in the geosciences as the availability of both measured and modelled data increases, and the careful
investigation of all possible physical interactions between the variables exceeds our capacities (Reichstein et al., 2019). For
post-wildfire debris-flow prediction, machine learning algorithms have in fact been shown to outperform logistic regression
models and ID thresholds in predictions (Kern et al., 2017; Nikolopoulos et al., 2018).

Here, we address two research questions. First, what is the uncertainty associated with estimation methods and with debris-
flow record length in ID-threshold parameters? By resampling the Illgraben debris-flow record using different time windows,
we estimate the confidence bounds of the ID-threshold parameters. Furthermore, we compare the uncertainties of two methods
that have been used recently for determining ID-threshold parameters (e.g. Leonarduzzi et al., 2017; Leonarduzzi and Molnar,
2020; Nikolopoulos et al., 2018). Second, how do traditional ID-threshold-curve methods compare with machine learning al-
gorithms? We extend the analysis of debris-flow prediction with additional rainfall event attributes in a random forest algorithm
(Breiman, 2001), test the predictive skill, and discuss the pros and cons of the multivariate approach for local debris-flow de-
tection based on different rainfall event properties (e.g. peak rainfall intensity, number of lightning events) and other seasonal
proxies, for example those related to sediment recharge.



## 2 The Illgraben study site

Illgraben is a north-facing catchment located in the Rhône valley in the Swiss canton of Valais. It consists of two subcatchments:
the eastern Illbach (4.15 km$^2$) is hydrologically and geomorphologically disconnected, while the western Illgraben (4.83 km$^2$) produces on average ~5 debris flows a year. The Illgraben sub-catchment has a maximum elevation of 2645 m a.s.l. at the summit of the Illhorn mountain. In this region, the main Rhône-Simplon fault line changes its orientation, resulting in numerous smaller faults in highly fractured bedrock and affecting the Illgraben catchment (McArdell and Sartori, 2021). The main sediment source area is a highly active hillslope underlain by quartzite bedrock, ranging from 1250 to 2370 m a.s.l. and with
slopes of up to 80°, where frequent landslides occur and deposit sediments in the trunk channel (Bennett et al., 2012; Berger et al., 2011). The main debris-flow channel starts just below this hillslope and is 5.2 km long. The first half is characterized by a mean slope of 16% until the fan apex at 886 m a.s.l. (Badoux et al., 2009). The second half is flatter (10%) and confined by check dams, before it joins the Rhône river at 605 m a.s.l.

In 1961 a large rock avalanche on the northern slope provided abundant sediment and increased the debris-flow frequency
in the following years (Hürlimann et al., 2003). However, since then this part of the catchment has produced sediment at much lower rates than the main source area (Schlunegger et al., 2009). The rock avalanche prompted the construction of 30 concrete check dams to stabilize the channel, with the most upstream one being 48 m tall (Lichtenhahn, 1971). This upstream check dam was effective in stabilizing the toe of the rock avalanche deposit and reducing the number of debris-flow events in subsequent years (Hürlimann et al., 2003).

Since 2000 the Swiss Federal Research Institute WSL has been operating an observation network in the Illgraben catchment, including rain gauges (added in 2001), geophones, depth sensors and a force plate (Rickenmann et al., 2001; Hürlimann et al., 2003; McArdell et al., 2007; McArdell, 2016). In this study, we use the rain gauge data and a debris-flow inventory including events up to the year 2017 (McArdell and Hirschberg, 2020). Illgraben has an alarm system that operates independently from the debris-flow observation station. It serves the villages of Susten and the Pletschen-subdivision on the eastern side of the
fan and protects the people visiting the Pfynwald nature park on the western side. Furthermore, hiking trails and sports fields close to the riparian zone make the fan a vulnerable area. Geophones mounted on check dams detect debris flows and activate alert lights and acoustic signals downstream in the riparian zone. After an initial phase of testing and optimization by WSL, the station was operated for about 10 years before the geophones and depth sensors described in Badoux et al. (2009) were replaced with sensors requiring less maintenance. The alarm system is now operated by the local municipality in cooperation
with an engineering company. For a detailed description of the original alarm system, the reader is referred to Badoux et al. (2009).

The mean annual precipitation at mean catchment elevation (1600 m a.s.l.) computed for the time period 1981–2010 was 900 mm y$^{-1}$ (HAD, 2015) and the mean annual temperature for the same period was 5.9°C (Hirschberg et al., 2021). Debris flows generally occur from May to October. Although climate change scenarios project longer debris-flow seasons in the future
(Hirschberg et al., 2021), and recently debris flows have also been recorded in April (2020) and December (2018), most debris flows occur in response to convective rainstorms between June and August. Snowmelt likely plays a role in spring but has never


**Figure 1.** Aerial view of the Illgraben study site located in the Swiss Rhône valley. Large amounts of sediment are generated on the active hillslope. Debris flows mostly initiate in the channel below this slope, and volumes are calculated based on data collected at the location of the force plate (source: Federal Office of Topography).

been observed to be the sole trigger for the recorded debris flows. Debris flows overflowing the banks are only expected in the case of the failure of a landslide-generated dam, levee failing or breaching, or if the channel conveyance capacity is reduced. Such a large debris flow was only observed after the failure of the dam formed by the 1961 landslide (Badoux et al., 2009).



## 3 Methods

### 3.1 Data

The timing and volume of debris flows were derived from the local observation system and are reported in McArdell and Hirschberg (2020). The debris-flow data set contains 75 entries from the years 2000 to 2017, with 1–8 debris flows occurring every year and bulk volumes ranging from 2000 to over 100,000 m$^3$ (median 25,000 m$^3$). The debris-flow mean bulk density was typically 1800–2200 kg m$^{-3}$ (Schlunegger et al., 2009). Because the local rain gauge (Fig. 1) was installed in June 2001, only 67 of these debris flows overlap with the rainfall record and were used in this study. The local rain gauge is a 0.2 mm resolution tipping bucket with a 10-min sampling rate. It is not heated and therefore only measurements in the period from May to October are considered, which coincides well with the debris-flow season. During this period, 480 mm of rainfall is measured annually on average, corresponding to about half of the total annual precipitation. The majority of debris flows are triggered by events with cumulative rainfall exceeding 5 mm. Although this amount is exceeded by about 30 storms each year, only 4.2 debris flows are triggered annually on average. Temperature data was provided by the Swiss Meteorological Office (MeteoSwiss) from the Montana meteorological station, located about 11 km northwest of the study area. The measurements were interpolated using local lapse rates to account for the difference in elevation of about 200 m, as described in Hirschberg et al. (2021). The 10 min total of recorded lightning strikes at a distance of 3–30 km was also derived from the Montana station and used as a secondary variable for the convective character of storms (Gaál et al., 2014) in the machine learning algorithm. The local predictive power of debris flows in Illgraben was also compared with a regional prediction of slope failure using a regional data set on shallow landslides in Switzerland including associated rainfall events (Leonarduzzi et al., 2017). It is based on a gridded daily rainfall product (RhiresD) and the *Swiss flood and landslide damage database* (WSL). This regional data set consists of 2137 landslides which occurred in Switzerland between 1972 and 2018 and for which damage was reported. Only the data from 2001 to 2017 was used, to be consistent with the local data set.

### 3.2 Performance statistics for debris-flow prediction

Confusion matrix and receiver operating characteristic (ROC) curves were used to compare models and optimize thresholds for debris-flow triggering. A confusion matrix can be computed for any binary classifier by counting the true positives (TP), false negatives (FN), true negatives (TN) and false positives (FP). Various performance statistics can be calculated from the confusion matrix. The most common measures in debris-flow and landslide forecasting are the following (e.g. Staley et al., 2013; Gariano et al., 2015; Leonarduzzi et al., 2017; Leonarduzzi and Molnar, 2020; Mirus et al., 2018):

Sensitivity (hit rate or true positive rate): $\quad SE = \dfrac{TP}{TP + FN}$ $\hfill (1)$

Specificity (complement of false positive rate): $\quad SP = 1 - \dfrac{FP}{FP + TN}$ $\hfill (2)$






True skill statistic (Peirce skill score or Hannsen and Kujpers discriminant:)   $TSS = SE + SP - 1$ (3)

Threat score (critical success index):   $TS = \dfrac{TP}{TP + FN + FP}$ (4)

The benefit of using the specificity over the false positive rate ($FPR = FP/(FP + TN)$) is that in a perfect model TSS,
sensitivity and specificity all equal 1. As noted by others (e.g. Leonarduzzi et al., 2017; Postance et al., 2018; Mirus et al.,
2018), optimizing TS leads to more conservative (higher) thresholds, while optimizing TSS yields more balanced rainfall
thresholds. The choice of score used in practice is therefore the user's decision. In this study, TSS was optimized to calibrate
thresholds and to compare classifiers, mainly because it is less sensitive to data sets with unbalanced class prevalence. In
particular, TSS was used in the following analyses:

1. In the determination of thresholds for single predictors (section 3.3)

2. In the determination of the ID-threshold parameters (section 3.4)

3. In the determination of probability thresholds from the random forest classifier (section 3.5)

4. In the comparison of these predictive models

Another metric for predictive model comparison is the area under the ROC curve (AUC). To estimate the AUC, sensitivity and
1-specificity are calculated and plotted for all possible threshold values. AUC equals 1 if there is a threshold that can perfectly
separate triggering and non-triggering events. A model with an AUC of 0.5 has no predictive power.

### 3.3   Rainfall event definition and other properties

The precipitation time series was discretized into rainfall events, which were separated by a minimum inter-event time (MIT).
Rainfall events were considered independent if no rainfall was recorded during this time. MIT is often chosen subjectively and
varies between 10 min at Chalk Cliffs (USA) to 6 h at Moscardo (Italy) for local debris-flow analyses (?Deganutti et al., 2000).
In Switzerland, a MIT of 2–3 h has been shown to be appropriate for the separation of thunderstorms (Gaál et al., 2014) and
storms initiating bedload transport in an Alpine watershed (Badoux et al., 2012). However, as the subjectivity in the rainfall
event definition complicates the comparison of rainfall thresholds, we followed the suggestion of Bel et al. (2017) to choose
MIT as the duration where the number of rainfall events stabilizes, which indicates its independence from MIT duration. In
this process, the sensitivity of rainfall threshold performance scores to the choice of MIT was also evaluated (Fig. 2a). The
measures SE, SP and TSS from fitting ID curves were found to decrease with increasing MIT as the number of events drops.
However, in absolute terms, the number of false alarms is higher at a short MIT because of the large total number of rainfall
events, and this also reflects in the TS statistic. At MIT = 3 h, the number of rainfall events stabilizes. This is seen in the stable
TS, meaning that there are no additional false alarms. The shape parameter of the ID threshold ($\beta$) also stabilizes at this MIT



**Table 1.** List of models and single predictors evaluated on their predictive performance for debris-flow triggering. Data source "Illgraben" refers to the local rain gauge, "Montana" to the MeteoSwiss weather station located 11 km from the catchment, and "Debris flows" to the Illgraben debris flow data set.

| Model name | Description |
| --- | --- |
| ID LR&TSS | Intensity-duration threshold where the shape parameter was determined by linear regression on the triggering events, and the scale parameter was set to maximize TSS |
| ID TSS&TSS | Intensity-duration threshold where both parameters were set to maximize TSS |
| RF_ID | Random forest model with rainfall event duration and mean intensity as input |
| RF_all | Random forest model with all single predictors as input |
| RF_ID+1 | RF_ID and one additional input (best single predictor, i.e. highest AUC) |
| RF_ID+var | RF_ID with four additional inputs: best Rmax, best Rant, best Ta and lightning strikes |

| Single predictor name | Description | Data source |
| --- | --- | --- |
| Duration | Rainfall event duration in hours | Illgraben |
| Rmean | Mean rainfall intensity in mm | Illgraben |
| Rtot | Total event rainfall in mm | Illgraben |
| RmaxTm | Maximum rainfall accumulation in T minutes within the event in mm ($10 \leq T \leq 120$) | Illgraben |
| RantTd | Antecedent rainfall within T days prior to the rainfall event in mm ($3 \leq T \leq 90$) | Illgraben |
| Tamin, Tamean, Tamax | Minimum, mean and maximum air temperature at the day of the rainfall event in °C | Montana |
| Taspan | Tamax - Tamin | Montana |
| lightings | Number of lightning strikes recorded within a radius of 3–30 km from the station | Montana |
| Freezing Days | The number of freezing days in the winter prior to the debris-flow season | Montana |
| time w/o DF | Elapsed time since the last debris flow in days | Debris flows |
| month, dayofyear | The month and the day of the year of the rainfall event | Debris flows |

(Fig. 2b). The scale parameter ($\alpha$) increases because the rainfall events become longer with increasing MIT and therefore the rainfall amounts increase. Consequently, MIT = 3 h was used throughout this study and it was confirmed that in the Alps an MIT of 2–3 h is an appropriate time period for separating rainfall into independent events for various applications, as found in independent studies by Badoux et al. (2012), Gaál et al. (2014) and Bel et al. (2017). Nevertheless, a suitable MIT should be objectively tested for each study site if possible.

Once MIT was defined, other rainfall-event properties were extracted in addition to the duration and mean intensity (Table 1, single predictors). Maximum cumulative rainfall was computed for accumulation periods of 10 min to 2 h. Antecedent rainfall was defined as cumulative rainfall from 3 to 90 days prior to the rainfall event. These event properties can be computed from

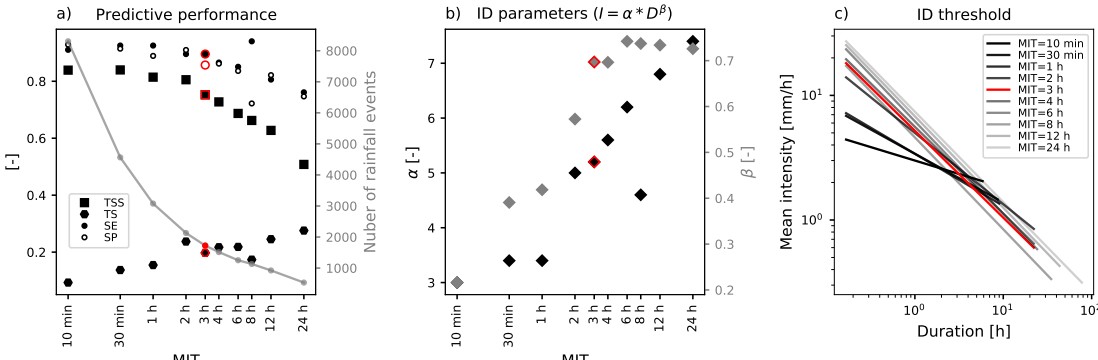

**Figure 2.** ID thresholds and their predictive performance statistics as a function of rainfall event definition (minimum inter-event time, MIT). (a) Sensitivity of true skill statistic (TSS), threat score (TS), sensitivity (SE), and specificity (SP) to MIT. The number of rainfall events varies as a function of MIT and is shown in grey. (b) Sensitivity of ID-threshold parameters to MIT. (c) Visualization of ID thresholds for different values of MIT. MIT = 3 h (red) was chosen for this study (see text).

any rainfall time series and are important for the soil moisture saturation and the likelihood of runoff formation in response to rainfall.

Furthermore, event properties related to air temperature were added. The daily mean, minimum and maximum temperature was computed for the day of each rainfall event. If an event spanned several days, the day the event started was considered. In the case of low temperatures, it could be snowing in the higher parts of the catchment, reducing the amount of liquid water contributing to subsequent runoff. As an indicator of convective rainfall, the daily temperature span and the number of lightning strikes were added. To account for seasonality effects, the day-of-year and the month of each event were included. As proxies

for sediment availability, the time elapsed since the last debris flow was added and the number of freezing days of the current hydrological year (November–October) was computed from the temperature time series. The aim of the latter was to account for sediment recharge related to frost-weathering processes (Hirschberg et al., 2021).

### 3.4   Rainfall ID thresholds

The best way to determine the scale ($\alpha$) and shape ($\beta$) parameters of rainfall ID curves and their uncertainties is an ongoing

discussion. Brunetti et al. (2010) presented a statistical (frequentist) approach involving estimating $\beta$ with a linear regression (in logarithmic space) fitted to all triggering rainfall events and decreasing $\alpha$ by an amount which is equal to the distance of the median residual to a chosen lower percentile. While this method is objective and, when applied as an EWS, makes it possible to control the hit rate, it neglects the information from the non-triggering rainfall events. Lately, confusion matrix and ROC methods (see section 3.2) have been used as objective measures to determine threshold parameters and compare the predictive

performance of different models.




For the ID thresholds computed here, sensitivity (Eq. 1) and specificity (Eq. 2) were calculated. The threshold performance was then evaluated in terms of TSS (Eq. 3). Two approaches were applied to optimize the ID-threshold parameters. In the first approach, as in the frequentist method, the shape parameter $\beta$ is determined in the log-log space with a linear least-squares approximation of the debris-flow triggering ID pairs. In a next step, the scale parameter $\alpha$ is tuned to maximize

TSS. This method is called LR&TSS hereafter. In the second approach, the scale parameter $\alpha$ and the shape parameter $\beta$ are simultaneously tuned to maximize TSS. This approach is hereafter referred to as TSS&TSS (Table 1, models).

To test the sensitivity of these two methods to the record length, resampled (bootstrapped) time series of rainfall and debris-flow events from 1 to 30 years were produced. Thus, only entire years were resampled, to avoid breaking up any natural intra-annual patterns. One sampled year consisted of all debris-flow triggering and non-triggering rainfall events. For example,

for a record length of 5 years, 5 annual samples were drawn with replacement from the 17-year observation period. This means that a specific year could occur multiple times in one 5-year sample. This procedure was repeated 100 times for each record length. Finally, the bias in ID-threshold parameters was estimated for each sample. The bias was defined as the relative deviation of estimates of $\alpha$ and $\beta$ from the corresponding reference values, i.e. the ones calculated from the original record using LR&TSS and TSS&TSS.

### 3.5 Random forests for debris-flow prediction

Much of the information contained in the rainfall time series, such as antecedent rainfall and peak intensity, is lost when discretized into events and characterized only by mean rainfall intensity and duration. As an alternative, random forests (RF, Breiman, 2001) were used to include more rainfall event properties (Table 1) for the classification of rainfall events into debris-flow triggering and non-triggering. Random forests are based on a statistical learning algorithm that uses multiple decision

trees. Each of these trees is trained with a subset of the predictor variables in the training data set. This procedure (also called bagging) is fundamental to the algorithm because it decreases the correlation among the trees and makes random forests suitable for capturing complex interactions and structures in the data. For detailed information, the reader is referred to Breiman (2001) and Hastie et al. (2009). The Scikit-learn module in Python was used to develop a random forest classifier (Pedregosa et al., 2011).

For the prediction of debris flows, logistic regression models have been used extensively in regional post-fire debris-flow studies to account for rainfall threshold variability due to spatial differences in slope and burned area (e.g. Cannon et al., 2010; Staley et al., 2017). Moreover, Bel et al. (2017) showed, for a French debris-flow torrent, that when ID thresholds were used in conjunction with a logistic regression model including variables for peak rainfall intensity, antecedent rainfall conditions and the number of days since winter, the number of false alarms could be reduced. RF was used instead in the present study,

because of its ability to consider multiple predictor variables with non-linear relationships and correlating predictors. To our knowledge, the predictive power of RF has not yet been tested for local debris-flow or landslide predictions. Nikolopoulos et al. (2018) used RF for regional post-fire debris-flow predictions in the western United States and showed that RF improved debris-flow predictions when a variable representing the location was added.





Here, four RF models – with the number of predictor variables ranging from 2 to 24 – were tested, with the first being the
equivalent of the traditional ID threshold (RF_ID, Table 1). The model output included the probability of being debris-flow
triggering for every rainfall event. The probability threshold for classification had to be tuned because the threshold-optimizing
TSS is likely not 50% but rather somewhat smaller (Nikolopoulos et al., 2018). The predictive performance was then compared
with the classical ID threshold, and with all single predictors (Table 1).

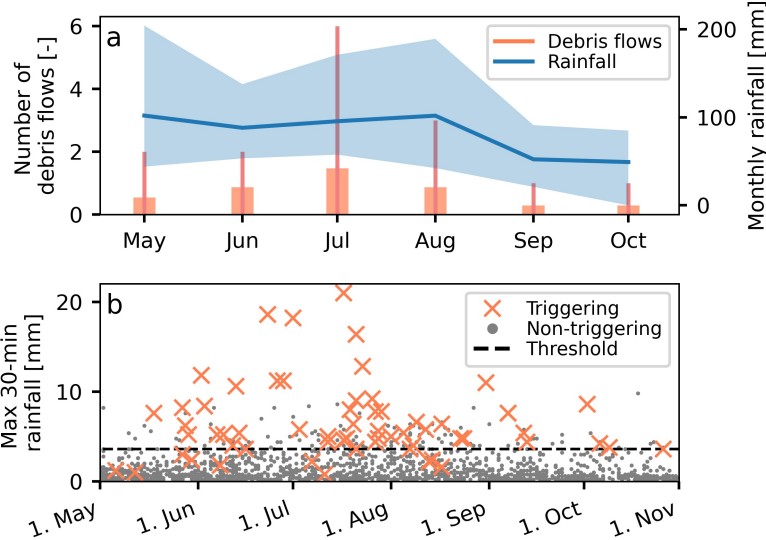

**Figure 3.** (a) Mean number of debris flows per month (orange bars) and mean monthly rainfall (blue line) in the Illgraben catchment. The thin orange lines and the blue shaded area show the inter-annual variability, i.e. the observed minimum and maximum in each month between 2001 and 2017. (b) Maximum 30-min rainfall accumulation in debris-flow triggering and non-triggering rainfall events. The threshold (3.6 mm 30 $\mathrm{min}^{-1}$, corresponding to 7.2 mm $\mathrm{h}^{-1}$) was determined by optimizing the TSS (0.75).

## 4 Results & Discussion

### 4.1 Debris-flow ID thresholds and their seasonality

In the study period from 2001 to 2017, 21 debris flows were triggered in spring and early summer (May and June, with the local influence of snowmelt), 38 in summer (July and August) and 8 in autumn (September and October) (Fig. 3). The monthly inter-annual variability was especially high in July, when between 0 and 6 debris flows occurred. This was also the month with the most extreme 30-min rainfall due to convective storms. The debris-flow activity dropped considerably in autumn, when the monthly rainfall also reduced by about 50%. Such seasonality is typical for Alpine debris-flow torrents (e.g. Schneuwly-Bollschweiler and Stoffel, 2012; Bel et al., 2017). In spring, snowmelt generates additional runoff and saturates the debris, which may lower the rainfall threshold for debris-flow initiation. This likely played a role in the events which were triggered at low rainfall amounts (30-min duration) before mid-June. There were also some lower-intensity events in July and August which still triggered debris flows. However, at this time of the year, inaccurate rainfall measurements and high spatial variability during convective storms are a more likely explanation. The most intensive 30-min rainfall event that did not trigger a debris flow was in October, possibly indicating sediment supply-limited conditions.

Debris-flow triggering threshold curves in Illgraben showed the typical negative power-law relationship between mean intensity and duration (Fig. 4). Debris flows occurred mostly during high rainfall intensities. However, triggering and non-triggering




events could not be separated perfectly. There were a few outliers at very short (<1 h) and very long rainfall durations (>16 h),
which were triggered by comparably little rainfall. ID thresholds were computed with two methods (TSS&TSS and LR&TSS)
for the entire data set and for individual seasons (Fig. 4). The scale parameter $\alpha$ had values between 2.6 and 7.3, with lower
values for TSS&TSS (2.6–5.4) and higher values for LR&TSS (5.2–7.3). The shape parameter $\beta$ was consistently smaller for
TSS&TSS (0.26–0.93) than for LR&TSS (0.52–0.94) and varied considerably between the seasons. Only in spring, the thresh-
olds were practically identical (Fig. 4b). For the entire data set, this resulted in the median TSS&TSS threshold being lower for
short durations ($\leq$4.5 h) and higher for long durations. The LR&TSS threshold was very similar to a curve defined earlier for
Illgraben, with $\alpha = 5.4$ and $\beta = 0.79$ (McArdell and Badoux, 2007), although there $\alpha$ was set to detect all triggering events. If
the same procedure had been used for the data set used here, the ID threshold would also be lower.

The seasonal ID thresholds differed, but direct comparison is difficult because of differences in the number of events (Fig.
4b–c). In autumn, it is clearer than in other seasons that only high-intensity rainfall events triggered debris flows. This may
be due to (a) rainfall measurements being more representative and accurate for the initiation area than in summer when more
convective events take place; (b) sediment availability being exhausted at the end of the debris-flow season (Berger et al.,
2011; Bennett et al., 2014); or (c) grain size coarsening throughout the wet season, increasing the hydraulic conductivity in the
channel bed and therefore also the rainfall threshold that must be exceeded to generate runoff (Domènech et al., 2019). As a
consequence, the false alarm rate was lower in autumn than in other seasons.

For longer durations, larger rainfall amounts are required for debris-flow triggering, and this reflects the balance of infil-
tration, storage and drainage of water. However, for short and long rainfall durations ID pairs fail to plausibly describe the
hydrological processes leading to landslide initiation (Bogaard and Greco, 2018). However, here it was not clear if this was
also the reason for the outliers triggered by lower mean rainfall intensities. There were two debris flows in summer at rainfall
durations of 10 and 30 min which were triggered by significantly lower mean rainfall intensities than the other debris-flow
events associated with these durations. One possible reason is that rain gauges, although close to the initiation area ($\sim$1 km),
are prone to not capturing peak intensities, especially of convective storms, even at short distances (Nikolopoulos et al., 2014;
Marra et al., 2016). These two events were also characterized by high antecedent rainfall, however, which could have lowered
the triggering threshold. In spring, three events were triggered at low mean rainfall intensities (<1 mm h$^{-1}$) and after more than
16 h. It could be that the MIT parameter does not separate rainfall events accurately in these cases, that the rainfall threshold
was lower due to snowmelt, that there were errors in the rainfall data, or simply that these debris flows were triggered by other
mechanisms than rainfall excess, such as the breaching of a small landslide dam.

Debris flows occur at a wide range of 14-day antecedent rainfall conditions, with many events occurring with very low
values for this variable (Fig. 4a). Antecedent rainfall does not appear to be a significant precondition for debris-flow triggering
in the Illgraben catchment. This has also been observed at other alpine locations (e.g. Abancó et al., 2016). Here, the debris-
flow magnitudes were not affected by the intensity of the triggering rainfall, as observed in other studies (Hirschberg et al.,
2019; Pastorello et al., 2018). However, the magnitudes were affected by the amount of antecedent rainfall. Higher antecedent
rainfall amounts lead to a higher degree of pore saturation along the entire channel bed. Sediment entrainment then experiences



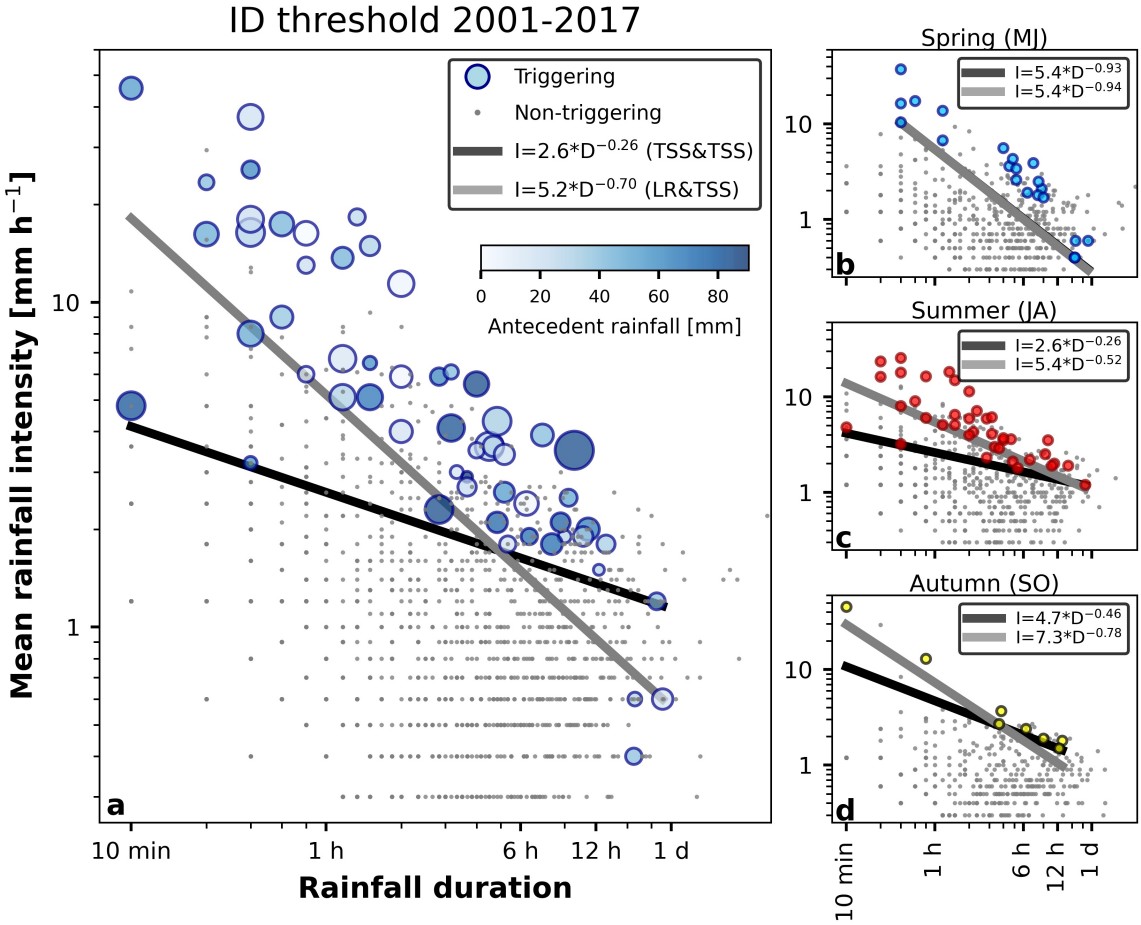

**Figure 4.** ID thresholds for the Illgraben catchment computed with two methods. (a) ID thresholds for the entire data set. For the 67 triggering events, the 14-day antecedent rainfall (blue colour) and the (relative) bulk volume (marker size) of the debris flows are also shown. (b)–(d) ID thresholds for each season, where the lines are the thresholds computed using TSS&TSS (black) and LR&TSS (grey).

a positive feedback from increased pore water pressure as the debris-flow surge passes by, increasing the debris-flow volume (Iverson et al., 2011; McCoy et al., 2012; Hirschberg et al., 2019).

## 4.2 Sensitivity of ID thresholds to record length and identification method

Illgraben ID thresholds where the ID-threshold parameters are jointly tuned to optimize TSS (TSS&TSS) have slightly larger TSS values, but are accompanied by higher uncertainties, measured as the $10^{th}$ to $90^{th}$ percentile range, in both $\alpha$ and $\beta$ biases, compared with ID thresholds with parameters estimated by first fitting $\beta$ to the triggering events and then tuning $\alpha$

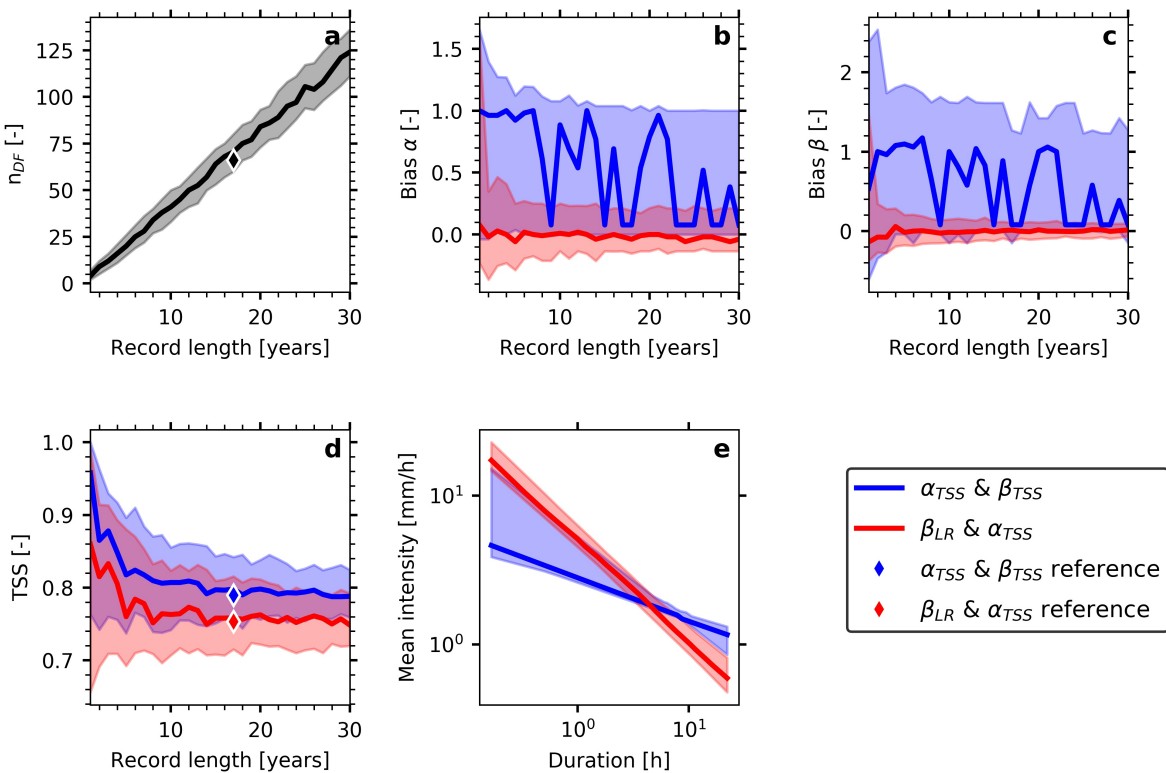

**Figure 5.** ID threshold sensitivity for the Illgraben catchment, assessed by resampling (bootstrapping) 100 times for each record length (1–30 years) from the 17–year data set of debris-flow triggering and non-triggering rainfall events. (a) number of debris flows; (b) bias in the scale parameter $\alpha$; (c) bias in the shape parameter $\beta$; (d) True Skill Statistic (TSS); (e) ID thresholds with variability from the observed record length of 17 years. Solid lines are the medians and shaded areas comprise the $10^{th}$ to $90^{th}$ percentiles of the resampled data. The diamonds (reference) refer to the values obtained when the full 17-year data set was used.

to optimize TSS (LR&TSS) (Fig. 5b–d). TSS&TSS thresholds are lower for short durations (<4.5 h) and higher for long
durations. TSS&TSS parameter estimates are overestimated by >100% even after 30 years of observations and the medians do
not converge (Fig. 5b,c). The fluctuations in the median suggest that there are 2–3 parameter sets with TSS values which are all
close to the optimum. Therefore, there are no unique ID-threshold parameters for Illgraben when estimated with TSS&TSS.
The uncertainty range is not located around zero but biased towards positive values because the reference values (see Fig.
4a, solid threshold line) are at the lower end of possible solutions. The medians and the uncertainty bounds from parameters
estimated with LR&TSS still converge after the reference record length of 17 years, with biases of ±20% for both $\alpha$ and $\beta$ (Fig.
5b,c). However, the biases decrease to ±30% already after 6 years or ∼25 triggering events. Furthermore, the TSS score, which





is overestimated for short records because it seems to be easier to fit an ID curve to only a few points, decreases and stabilizes after 6 years. Because the Illgraben record is a local data set, it is not affected by climatic, topographic, lithologic or land-use differences between sites as present in most large-scale regional data sets. Therefore, the uncertainties can be associated with

the respective method with high confidence and the Illgraben data set is well-suited to study such ID-threshold sensitivities.

Important advantages of using TSS for ID-threshold parameter estimation are that information from both triggering and non-triggering events is considered with equal weight and that the measure is prevalence independent. However, although the narrow uncertainty range for long record lengths (Fig. 5d) additionally indicates the robustness of the TSS score, it does not imply robustness in the parameter estimates that are based on TSS (Fig. 5b,c). In our case, with 67 debris flows, the ID-

threshold parameters computed with TSS&TSS seem to be highly sensitive to a few triggering events, which may be outliers but exist in any data set and are difficult to single out with certainty. Consequently, for local ID thresholds we advise against simultaneously optimizing $\alpha$ and $\beta$ against the TSS score (TSS&TSS). Local data sets are often comparatively small (<100 triggering events), and therefore this method can be sensitive to outliers.

Conducting the same sensitivity analysis on the regional data set, characterized by many more triggering events ($\sim$800),

which are mostly shallow landslides in this case, low prevalence (0.05%) and different triggering locations, showed the opposite result than for the local data set. With the regional data set, TSS values can be slightly enhanced when using the TSS&TSS instead of the LR&TSS method (Fig. 6d), as in the local data set. In contrast to the local data set, this enhancement in TSS is not accompanied by larger uncertainties in ID-threshold parameters (Fig. 6b,c). LR&TSS thresholds are practically flat (Fig. 6e) and parameter ranges are still converging after the reference record length of 17 years. Using LR&TSS even makes the

duration redundant because $\beta$ estimated with LR&TSS converges to 0. Note that the large range in $\beta$ bias for LR&TSS (Fig. 6c) is also because $\beta$ is close to 0, and absolute values are small (see section 3.4). TSS&TSS threshold parameters converge after $\sim$8 years, corresponding to $\sim$400 landslides. This is more events than reported for data sets in Italy, where $\sim$200 was enough both on the regional and on the national scale (Peruccacci et al., 2012, 2017).

The regional data set is inherently subject to much larger uncertainties, which can be disregarded in the local data set. These

uncertainties are mainly related to climatic, topographic and lithologic differences among the landslide locations. These differences may also lead to the slope of the ID curve, when fitted with linear regression, losing the typical power-law relationship of extreme rainfall. Furthermore, rainfall uncertainties are higher because regional analyses rely on interpolated precipitation (Frei and Schär, 1998). TSS&TSS instead profits more from the information in the non-triggering rainfall, by setting a threshold high enough to be above the many non-triggering rainfall events with low intensity and short duration, and steep enough to

detect triggering events as a response to long-lasting rainfall (Fig. 6e).

The main differences between the methods used here and the well-known frequentist method (Brunetti et al., 2010) are that non-triggering rainfall events are considered either in the determination of the scale parameter $\alpha$ (LR&TSS) or in both $\alpha$ and the shape parameter $\beta$ (TSS&TSS). Parameters estimated by LR&TSS for the local data set have lower uncertainties than when estimated by TSS&TSS. For the regional data set the TSS&TSS method yields both better predictions and lower parameter

uncertainty. Hence, this comparison of a local and a regional data set suggests that the range of climatic, topographic, lithologic or land-use differences within a data set should be considered when deciding which method to apply for ID-curve estimation.

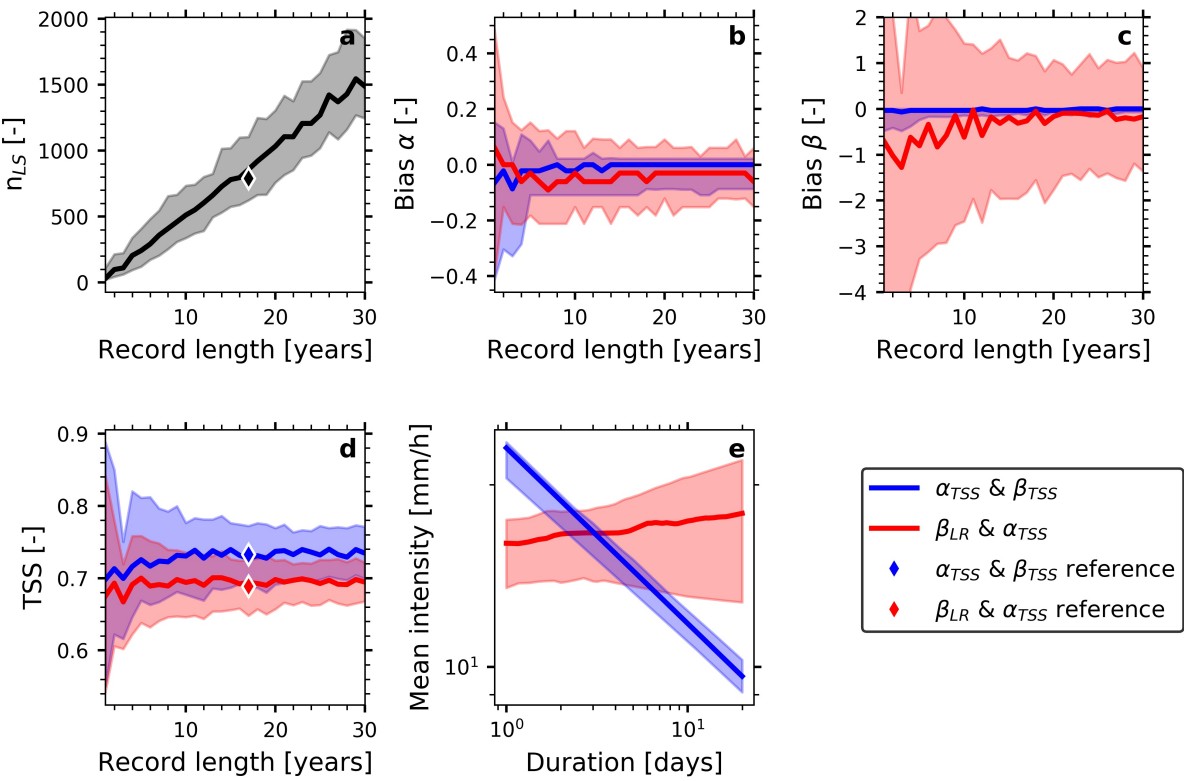

**Figure 6.** ID-threshold sensitivity for the regional data set, assessed by resampling (bootstrapping) 100 times for each record length (1–30 years) from the 17-year data set of landslide triggering and non-triggering rainfall events. (a) number of landslides; (b) bias in the scale parameter $\alpha$; (c) bias in the shape parameter $\beta$; (d) True Skill Statistic (TSS); (e) ID thresholds with variability from the observed record length of 17 years. Solid lines are the medians and shaded areas comprise the $10^{th}$ to $90^{th}$ percentiles of the resampled data. The diamonds (reference) refer to the values obtained when the full regional data set is used.

## 4.3 Predictive power of uni- and multivariate models

We compared debris-flow triggering proxies related to maximum precipitation intensity, antecedent rainfall and temperature, among others (Table 1, lower part). Each proxy was evaluated as a single predictor in terms of TSS. These were also compared
with five multivariate models: the LR&TSS-ID threshold and four RF classifiers with different single predictors as input (Table 1, upper part). The TSS&TSS-ID thresholds were excluded here due to the large uncertainties in estimated parameters. The models were validated with five-fold cross-validation (CV).

We found that the RF classifier's TSS values (0.77–0.81) are only slightly higher than the classical ID threshold's TSS (0.76) (Fig. 7a). This improvement is due to a generally higher sensitivity score (Fig. 7b). However, as seen in the distribution of the

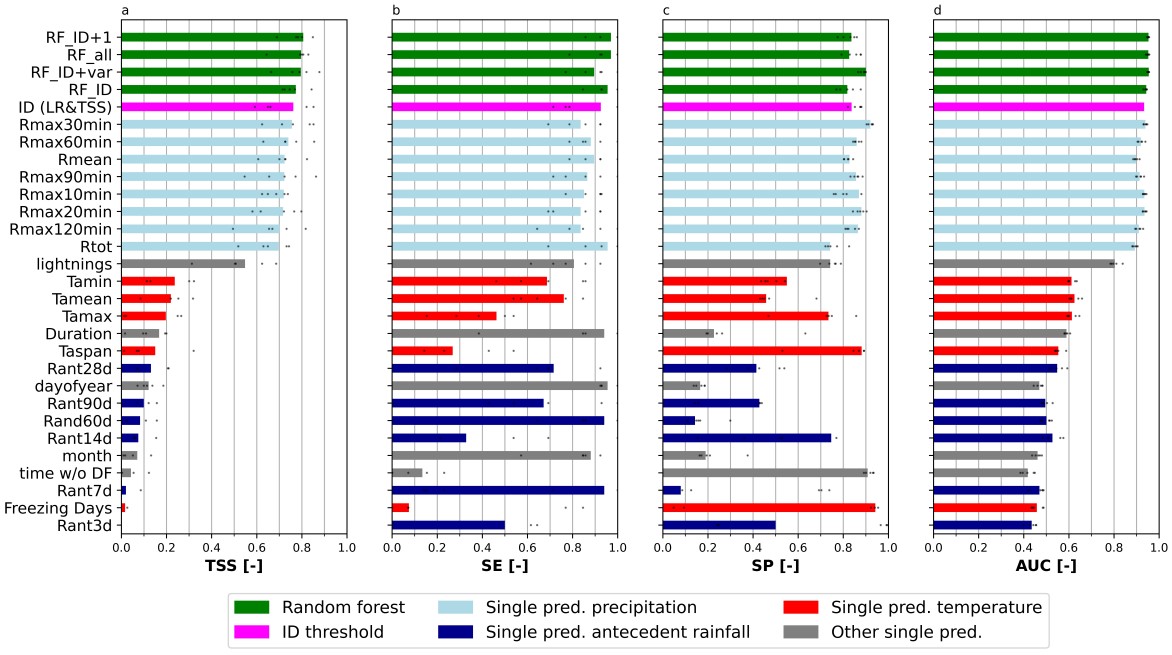

**Figure 7.** Model comparison for the prediction of debris flows in the Illgraben catchment, with reference to (a) true skill statistic (TSS), (b) sensitivity (SE), (c) specificity (SP) and (d) area under the curve (AUC). The model descriptions are given in Table 1. The bars refer to models using the full data set for training. The grey dots represent results from the five-fold cross-validation.

points from the CV (small grey points in Fig. 7), RF can reduce uncertainties. The best improvement in TSS is seen when only one additional predictor (max 30-min rainfall) is added to the intensity and duration of rainfall events. If all available data are used by an RF classifier (RF_all), the performance even decreases compared with RF_ID+1. This is likely due to overfitting and can happen when many of the input variables have poor predictive power. In this case, the RF classifier is fitted to the noise from the poor predictors while the predictive information from other variables gets lost. Hence, CV avoids the overfitting of a model set-up, and further improvements may be achieved by testing different combinations of variables.

This analysis demonstrated that the best single predictor in terms of TSS is the 30-min max rainfall (TSS=0.76), and its specificity is also the highest overall (0.92) (Fig. 7c). In general, the predictors relating to max rainfall event intensities of different time scales have relevant predictive power, while predictors related to antecedent rainfall do not. Of all debris flows, 96% can be identified with a threshold (5 mm) for total event rainfall (Fig. 7b). Furthermore, 30-min maximum rainfall and mean rainfall intensity on their own perform similarly to the ID threshold. These rainfall properties are simple to calculate with a moving window from any rainfall time series, even without adding uncertainty by rainfall event separation. Surprisingly, lightning strikes also have some predictive power, even though they were recorded within an area which is almost 600 times



larger than the catchment size ($\sim$2800 km$^2$). This indicates that debris flows are favourably triggered when thunderstorm clusters occur. The single predictors listed below lightning strikes in Fig. 7 have (almost) no predictive power because their
AUC values are $\sim$0.5, which is close to a random guess.

Although ID thresholds are widely used in applications, a common problem is the number of false alarms they cause. For Illgraben, although 92% of the debris flows are detected with the ID threshold estimated with LR&TSS, only 20% of the rainfall events exceeding the threshold are expected to produce a debris flow. To increase this accuracy mainly based on different rainfall properties, we systematically analyzed the predictive power of single predictors and multivariate models based on the random
forest algorithm. Although the RF classifier only marginally improved the TSS score, the potential of overcoming some of the well-known limitations of ID thresholds is evident. For example, rainfall properties could be combined with measured, remotely sensed or modelled variables, such as discharge or soil moisture products (Wicki et al., 2020). Recently, random forests have been used to detect mass movements from seismic signals (Wenner et al., 2021; Chmiel et al., 2021) and could be coupled with rainfall measurements or forecasts to potentially increase the accuracy. It would also be interesting to study
the spatio-temporal rainfall structure from radar-based rainfall estimates and their influence on debris-flow triggering (Marra et al., 2016). Of course, random forests are only one alternative to ID thresholds, and there are other algorithms to be tested (see Kern et al., 2017, for a review on post-wildfire debris flows). A drawback of such empirical thresholds is that long-term observations are required to establish them. Where such data are available, additional predictors can easily be implemented in a RF classifier, as presented here, and tested regarding their predictive power.





## 5 Conclusions

In this study we used a 17-year record of precipitation and debris-flow timing and magnitude to complete a systematic analysis of rainfall conditions leading to debris-flow triggering in a Swiss catchment, Illgraben. Based on 67 debris-flow triggering and 1657 non-triggering rainfall events (prevalence of 3.8%) we defined a rainfall intensity-duration threshold $I = 5.2 * D^{-0.70}$ with the most suitable fitting method applied in this work. Given the high debris-flow frequency in Illgraben, it can be considered as a lower threshold for rainfall-induced debris flows in the Swiss Rhône valley.

Debris-flow activity is greatest in summer, coinciding with peaks in total monthly rainfall accumulations and peak 30-min rainfall. Although we find differences in seasonal ID thresholds, they are partly based on the occurrence of only a few triggering events (e.g. in autumn). It remains challenging to determine if the reason for this seasonality is seasonal differences in rainfall or in sediment availability or processes such as snowmelt or grain size coarsening.

Our systematic analysis of the uncertainties associated with ID thresholds shows that, for a catchment with rainfall-induced debris flows, 25 debris-flow observations are sufficient to constrain the ID-threshold parameters $\alpha$ and $\beta$ with uncertainties of $\leq 30\%$. However, our findings demonstrate that this uncertainty strongly depends on the data set and the method used to determine ID-threshold parameters. When comparing the Illgraben (local) data set with a Swiss landslide (regional) data set, more triggering events (400) were needed for threshold parameters to converge in the regional data set, due to the higher spatial variability in the data set. More importantly, the best method to minimize uncertainties changed from LR&TSS for the local to TSS&TSS for the regional data set. This underlines the need for standardized methodologies for rainfall threshold identification and validation, and proper reporting of the methods used (see Segoni et al., 2018).

Finally, we aimed to lower the false alarm rate often associated with ID thresholds. Using a random forest model including the predictors rainfall event duration, mean rainfall intensity and the 30-min maximum rainfall amount increased the TSS (true skill statistic) by 0.04 (i.e. ∼3 more hits or ∼70 fewer false alarms). Adding more input variables to the random forest model, including antecedent rainfall, did not improve the performance. Although the expectation of significantly decreasing the false alarms was not fulfilled, we present a flexible framework where additional input variables can easily be tested. The aim of future work should be to include variables such as modelled or measured soil moisture or information on spatio-temporal rainfall structure from radar-based rainfall estimates. Machine learning algorithms can be helpful for maximizing information exploitation from available data and for increasing the accuracy of (debris-flow) early warning systems, and we have highlighted this potential.



*Data availability.* The debris-flow volumes are available from the Environmental Data Portal EnviDat (McArdell & Hirschberg, 2020, http://dx.doi.org/10.16904/envidat.173). Climate data are available for research purposes from the agencies mentioned in section 3.1.

*Author contributions.* JH conducted the analysis, produced the figures and wrote the original article draft. JH, BMA, AB and PM designed
the study. EL provided the metadata of the regional data set and guided the analysis. All authors contributed to the interpretation of the results and to the revision of the article.

*Competing interests.* The authors declare that they have no conflict of interest.

*Acknowledgements.* This study was partly funded by the WSL research program CCAMM (Climate Change Impacts on Alpine Mass Movements). PM acknowledges funding from the Swiss National Science Foundation (grant no. 165979). We thank MeteoSwiss for providing
climate data and the municipality of Leuk for providing rainfall data. We thank Melissa Dawes for improving the use of English in an earlier version of this manuscript.





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
