# Peer review of "Evaluating methods for debris-flow prediction based on rainfall in an Alpine catchment"

_Natural Hazards and Earth System Sciences, 2021_

## Author Comment (AC1)

Response to referee comment by Ben Mirus

We thank Ben Mirus for providing a constructive review and his interest in our study. Here we respond to the comments and address how we will implement the changes in a revised manuscript.

**General comments**

*There are quite a few minor details that are missing in the abstract, most of which could be gleaned eventually from reading the entire paper, but should be included in the abstract for completeness.*

We will adapt the abstract according to the answers to the specific comments below.

*…I could not find information on how storms and ID are defined for the regional datasets, nor did I find a clear explanation that multiple durations for antecedent rainfall conditions were explored in the RF model.*

We will add these specifications according to the answers to the specific comments below.

*…some further details on the rationale behind the scenarios selected for different optimization strategies would be helpful.*

We will add details on the rationale behind the scenarios according to the answers to the specific comments below.

**Specific comments**

*I found the title quite critical, when actually the paper is not only about limitations, but rather a comprehensive evaluation of multiple approaches to debris-flow forecasting. Perhaps the title could be revised to more fairly represent the important contributions of this work.*

We agree that the title is a bit unrewarding. We will think of a new title and for now propose "Evaluating methods for debris-flow prediction based on rainfall data in an Alpine catchment".

*L3. I'm not sure I agree that there are no standardized procedures. I think it's more reasonable to state that there are multiple competing methods that have not been objectively and thoroughly compared at multiple scales.*

We will adapt the sentence accordingly.

*L6. Consider stating "record duration" since you are talking about time, not a distance (length).*

That makes sense. We will make adaptations accordingly.

*L12. Regional landslide dataset with local rainfall input or with a regional rainfall database? This is a critical detail that needs to be clarified in the abstract even if it can be determined later in the paper.*

The regional landslide database is combined with a regional rainfall database, i.e. with a gridded interpolated daily rainfall from stations. We will specify this.

*L13. If these implications are important, is it important enough to list them in the abstract? Also, state here whether the RF model was tested for just local or also for regional?*

We left the reader in the dark here. We will add that the major implication is that the appropriate method depends on the available data set.
The RF model was only tested for the local case. We will add this information to the text.

*L15. I found this "30-min maximum accumulated rainfall" a bit confusing as it isn't really standard terminology. Is this the greatest accumulated depth of rainfall observed within a given 30-minute period of a storm? If so, wouldn't that be basically equivalent to the peak 30-minute rainfall intensity (I-30)?*

Yes, it is the peak 30-min rainfall intensity. We will change the term here and in the rest of the manuscript.

*L17. Increase in predictive performance over which other threshold optimization approach/approaches?*

RF yields a slight increase over the ID-threshold (LR&TSS). We will adapt the sentence.

*L41. Again, it's not that there are none, but that a few established procedures are in use and that those approaches have not been compared objectively and thoroughly.*

We will adapt the sentence accordingly.

*L82. Could also mention that you evaluate these differences for both a local vs. regional landslide inventory.*

Thanks, we will add it.

*Figure 1. Legend should explain what the blue shaded channel and also the X marks the Illhorn peak. It wouldn't hurt to put the elevation of the Illhorn and the force plate or catchment outlet to provide easy reference for the steepness of the basin.*

We will add these specifications.

*L143-146. Consider briefly explaining the gridded daily rainfall product, including the spatial resolution and how it is collected/calculated, as well as what rainfall value was used for the threshold evaluations (i.e., did you use rainfall values from the nearest grid cell, or some grid-cell averaging, or …?). This is important context for evaluating the ID thresholds at the regional scale vs. local scale.*

The rainfall dataset for the regional analysis consists of a 1km x 1km gridded product of daily precipitation sums obtained by interpolation of rain gauges (ca. 420), accounting for local climatology and precipitation-topography relationships. We will add these specifications.

*L173-202. I guess you didn't explain the regional data here in Table 1. Perhaps that's not necessary, but you do need to define your MIT for the daily/regional data analysis. How are multi-day storms determined?*

The MIT for daily rainfall is 1 day and multi-day storms consist of a sequence of wet days. This definition of events overestimates event duration and underestimates mean intensity compared to hourly data, but compensates for this by longer and more dense records (see Leonarduzzi and Molnar, 2020). We will add these specifications.

*L192. Initially, I assumed that this 3-90d antecedent conditions meant the cumulative rainfall total measured between 3 and 90 days prior to the storm event. While there needs to be some explanation of why 3 days was selected as a cutoff (why not 2 or 1 day?), there also needs to be a clearer explanation that multiple potential durations of antecedent rainfall were considered. This only becomes apparent in Figure 7 and the associated analysis of the RF results and variable importance.*

Your assumption is correct. We cut off at 3 days to reduce the number of variables and it didn't make any difference in the results. However, this certainly requires some clarification. We will adapt the paragraph accordingly.

*L208. Yes, and thus does not consider the rate of false alarms.*

We will add this clarification.

*L204-216. These paragraphs could benefit from an explanation of the shape parameters in terms of how they influence ID threshold shape/position, and then subsequently the rational for why the two contrasting optimization approaches (LR-TSS vs, TSS-TSS) were selected. It might not be clear to all readers the significance of these choices.*

We will add the ID equation so the reader can directly see where scale (α) and shape (β) parameters come into play. We'll add an explanation saying that in the log-log space α controls the intercept and β the slope of the threshold line. We will also stress that the two approaches mainly differ in the way β is obtained: TSS&TSS considers both triggering and non-triggering events while LR&TSS only considers triggering rainfall events. This is a fundamental difference.

*L223. Consider clarifying the "... original complete (or 17 year) record..."*

We'll add this clarification.

*L248. By "classical" ID thresholds, you mean those optimized with ROC statistics (LR-TSS, TSS-TSS)?*

Yes, we will clarify this.

*Figure 3. Difficult to see what the minimum number of debris flows are in each month, but it looks like they're all zero. If so, consider just stating in the figure caption. (b) also, see previous comment about maximum 30-min accumulation. Isn't this just more or less equivalent to the peak I-30 (i.e. 7.2mm/h)?*

We will improve the visibility of the number of debris flows and/or clarify it in the caption. Yes, it is peak I-30 and we will change the term (see also answer above).

*L256-257. If seasonal snowmelt is a relevant control on rainfall triggering, then the antecedent precipitation variable ought to somehow account for this, but I suspect it cannot.*

We do not have direct evidence that snowmelt is a relevant control on debris-flow triggering. Seasonality in snowmelt is of course present, but snow cover cumulates over a long period of time, so it is not straightforward to account for this as antecedent precipitation. We hoped that the thresholds in Fig. 4 would clarify this but the seasonal thresholds were not conclusive. We could only confidently account for antecedent rainfall. Accounting for antecedent wetness, including infiltration from snowmelt, would require hydrological modelling since we don't have any measured data on snow or soil moisture.

*L260-261. Again, these non-conforming observations might also be related to the fairly coarse consideration of antecedent rainfall.*

Thanks, we will adjust the sentence accordingly.

*L266-268. As a discussion point, it could be interesting to compare this range in parameter variation to the ranges of typical ID thresholds reported in the literature, say for example the difference between values for Caine vs. Guzzetti et al. ID thresholds. I have not done this comparison myself, but it could be worth looking at.*

Bel et al. (2017, Fig. 12) did a thorough comparison of ID thresholds for debris-flows torrents. We will refer to this study here.

*L372. Even though 20% seems low, this is actually pretty good performance overall for an ID threshold relative to others developed worldwide, so that just further highlights the multitude of complex interactions that lead to debris-flow triggering and justify the need to explore more data-rich approaches like the RF you propose.*

Thanks, we will add this observation to the discussion.

**References**

Leonarduzzi, E. and Molnar, P.: Deriving rainfall thresholds for landsliding at the regional scale: Daily and hourly resolutions, normalisation, and antecedent rainfall, Natural Hazards and Earth System Sciences, 20, 2905–2919, https://doi.org/10.5194/nhess-20-2905-2020, 2020.

---

## Author Comment (AC2)

Response to referee comment by Clàudia Abancó

We thank Clàudia Abancó for providing a constructive review and for her interest in our study. Here we respond to the comments and address how we will implement the changes in a revised manuscript.

**Specific comments:**

*Title: It may be a bit misleading. It does not deal with the limitations of the thresholds but more with the uncertainties on their definition? I would suggest reconsidering it...*

We agree that the title is a bit unrewarding. We will think of a new title and for now propose "Evaluating methods for debris-flow prediction based on rainfall data in an Alpine catchment".

*L65: I would suggest adding a few references of studies using different MIT, as it is said they range from 10 min to 6 h but no references are given (although they appear later, in 3.3., but I would add them here too)*

We will add references here, including the ones in section 3.3.

*L82: Although the two methods that are going to be compared are mentioned in the abstract, I would list them here too*

We will do that.

*L80-85: I miss here stating as an objective (maybe as a secondary one) the analysis of the performance with local vs. Regional dataset, which is stated in the abstract.*

Thank you for pointing this out. The local vs. regional analysis should be mentioned here. We will add that we investigate how the uncertainties associated with the methods used for a local data set with local rain gauges change when the same methods are applied to a regional data set with gridded rainfall information.

*L89-98: Cite Figure 1 in this paragraph*

We will do that.

*L106: Rain gaugeS in plural? If there's more than one, why in the Figure only 1 is shown? Why only data from 1 is used?*

There are two other gauges in the basin which are not suited for this study. One has been moved during the study period and the other is sheltered by trees. Furthermore, the used rain gauge is the one closest to the triggering area. We will add this explanation.

*L111-113: If the geophones and depth sensors have been removed, which sensors (less maintenance) have been installed?*

We created a bit of confusion here. Badoux et al. (2009) describe the alarm system and some of these sensors have been replaced. However, these are not the same sensors used for the debris-flow monitoring conducted by the research institute WSL, which this paper focuses on. Here, only information on the time of debris-flow occurrence was used as it was detected at the force plate or earlier by geophones upstream. We will add explanations on these sensors and indicate them in Figure 1.

*L115: By including the citation of Badoux in line 113 I think you could delete it in Line 115, it is clear for the Reader that details can be found there*

We will delete this last sentence as it is redundant.

*L122: Any reference where snowmelt has been observed (even if not as sole trigger)?*

We will add that although snowmelt in many places adds considerable amounts of liquid water to the debris (Mosterbauer et al., 2018), it has never been observed to be the sole trigger in Illgraben.

*Figure 1: Why only force plate is indicated? I would suggest adding the other sensors (the new ones replacing the geophones)?*

We will add the locations of the relevant sensors and label them (see also response to L111-113).

*L135: 5 mm? This sounds like a very low number...*

It does sound low but the Illgraben ID thresholds are also lower than in other places. Debris flows are either triggered in individual gullies or in the main channel. When the latter is the case, water drained from multiple gullies with low infiltration rates concentrates in the channel. This possibly explains the low thresholds.

*L136: Does the local rain gauge not have a thermometer? Also, I would suggest moving Temperature, lighning strikes and other parameters to another paragraph, as I understand these are all variables for the Machine learning, but not actually for the main ID thresholds comparison? I was a bit confused reading about rainfall and changing to temperature abruptly, as it's the first time you mention the temperature variable*

There is a thermometer at the local rain gauge but because the sensors are not properly shielded, the measurements are unreliable.
We will make two paragraphs to better separate variables only used for machine learning.

*L160: Reference of TSS?*

We will add references.

*L169: Include Area Under Curve as clarification of what AUC stands for*

We will do this.

*L173: I understand that you used the same criteria for both trigg and non trigg rainfalls?*

Yes, discretizing the time series into rainfall events was done before separating into trig. and non-trig., so we used the same criteria. We will specify this.

*L175: Delete ? before Deganutti*

LaTeX couldn't find the reference due to a typo. We will correct that.

*L180 and Figure 2: I think this is more results than methods?*

We decided to add it to the method part because it was a necessary step in the process of defining rainfall events and in the result section we wanted to focus on the results addressing our research questions. As mentioned in the text (L. 178), we followed Bel et al. (2017) for this purpose, who thoroughly discusses ID threshold sensitivity to MIT.

*Figure 2 (b): Sensitivity? This word may be confused by SE? Could it be called "Analysis of ID-threshold parameters with changing MIT"?*

Good point. We will follow your suggestion.

*L184: I would not say that β stabilizes, but reduced the increasing tren dat MIT 3h... (in Fig 2b)*

We will adapt the sentence accordingly.

*L197: Actually, if it was snowing the data from the rain gauge would not be valid, right? As it is not heated... Have you considered this? If so, maybe you could mention here.*

Yes, this is an additional reason not to trust the rainfall measurements when it was cold. We will add it here. However, most debris flows occur in summer when solid precipitation is rare even for the higher parts of the basin. Therefore, there are only very few data points where this is the case.

*L208: This last sentence of the paragraph ("Lately, confusion matrix...") is actually a bit confusing to me. You have not used frequentist method, right? You used linear squares and LR&TSS and TSS&TSS methods if I have understood right. Therefore, the sentence is confusing as it seems that you have calculated the confusion matrix and ROC for the frequentist method...*

In this paragraph we reflect on how ID threshold parameters are determined in literature. We will clarify that determining ID threshold parameters using confusion matrix and ROC, on which we rely on in this paper, is an alternative to the frequentist method.

*L220: This is also confusing. A record of length 5 years includes 5 annual samples that can include repetition of the same year? Why is the procedure repeated 100 time for each record length? Please clarify*

We agree that the sentence is a bit confusing. The procedure is resampling with replacing and is frequently used for uncertainty assessment. Without replacing, we wouldn't be able to estimate uncertainty bounds because for 17 years there would only be one possible combination. Thus, for each record duration (T), we repeat the sampling with replacement 100 times to have enough combinations of years (i.e. for each T there are 100 samples consisting of T years). The stable median and uncertainty bounds confirm that 100 repetitions were sufficient. We will clarify this and add a reference to the resampling method.

*L280: I think it would be good to see the total rainfall amounts at some point in a Figure, as it is stated here that long duration need more rainfall (logic, but still nice to see)*

We will add a panel to Fig. 3 with total rainfall amounts.

*L297: Higher antecedent rainfall amount may lead to higher degree of pore saturation along the entire channel bed, but also, in some cases the antecedent rainfall would mostly contribute to the generation of lateral flow and increase of water table (e.g.:*
*M.N. Papa, V. Medina, F. Ciervo, A. Bateman 2013, Derivation of critical rainfall thresholds for shallow landslides as a tool for debris flow early warning Systems). This could also correlate with the fact that magnitudes are bigger, but I would say that the correlation between the antecedent rainfall and the magnitude it is a tricky point and needs careful evaluation...*

Thank you for pointing this out. We will add it to the discussion. However, in Illgraben in Hirschberg et al. (2019) we found that 14 days antecedent rainfall best correlated with the magnitudes, and we also tested shorter durations. Our thinking is that in such a steep gully, the rainfall contributing to lateral flow certainly happens at shorter time scales which is closer to few hours rather than 14 days. This gives us confidence that antecedent rainfall influences the saturation in the channel bed, and therefore the debris-flow magnitudes.

*L305: TSS&TSS thresholds are lower for short durations (<4.5 h) and higher for long durations- after this I would add (Figure 5e)*

We will add it.

*L311: However, the biases decrease to _30% already after 6 years or _25 triggering events- For both? Or only for 6? I can't see it that clearly in alpha?*

For both. We will match the y-axes in order to make it clearer.

*L335: Also, the source of rainfall data is different, right? If I am not wrong the work of Leonarduzzi et al. it was not based only in rain gauge data. Therefore, apart from climàtic, topographic and lihologic uncertainties it may be also from the type of rainfall data?*

Yes, you are right. We will add a better explanation hereof the work in which different rainfall data sets and resolutions were tested (Leonarduzzi and Molnar, 2020). We will also add a better description of the regional dataset (as suggested by RC1). The rainfall dataset

for the regional analysis consists of a 1km x 1km gridded product of daily precipitation sums obtained by interpolation of rain gauges (ca. 420), accounting for local climatology and precipitation-topography relationships.

*Figure 7:*
- *The grey dots are very difficult to see, specially over blue, red and green bars. Change colour of bars or make dots bigger*
- *I find this figure particularly dense and a bit difficult to follow. Some ideas on how it could be made a bit easier to read:*
    - a) *I understand that RF_ID+1 is based in one sigle predictor (the one with best performance). Why not indicating which one instead of leaving the reader to interpret?*
    - b) *Same with RF_ID+var and 4 predictors*
    - c) *Maybe then it would not be necessary to include all the single predictors in the same figure. Either include them in a separate figure or as supplementary material?*
    - d) *If you think it is relevant to keep the same format, I would suggest indicating the selected predictors for each RF model in some way...*

We will follow your suggestion about the dots and the labels. Half of the figure is filled by predictors without predictive performance, so we will visually better differentiate between models and single predictors with predictive performance and the ones without. This should make the figure more accessible. We think it's relevant that all used predictors remain in the figure to directly see the best predictors without having to look for the irrelevant ones in the text.

**References**

Hirschberg, J., McArdell, B. W., Badoux, A., and Molnar, P.: Analysis of rainfall and runoff for debris flows at the Illgraben catchment, Switzerland, in: Debris-Flow Hazards Mitigation: Mechanics, Monitoring, Modeling, and Assessment - Proceedings of the 7th International Conference on Debris-Flow Hazards Mitigation, pp. 693–700, 2019.

Leonarduzzi, E. and Molnar, P.: Deriving rainfall thresholds for landsliding at the regional scale: Daily and hourly resolutions, normalisation, and antecedent rainfall, Natural Hazards and Earth System Sciences, 20, 2905–2919, https://doi.org/10.5194/nhess-20-2905-2020, 2020.

---

## Author Response (AR1)

Prof. Thomas Glade
Editor
NHESS
19.07.2021

Dear Prof, Glade,

Herewith, we resubmit our manuscript nhess-2021-135 entitled "Evaluating methods for debris-flow prediction based on rainfall in an Alpine catchment". We are grateful to Ben Mirus and Clàudia Abancó for their constructive comments, which helped to improve the manuscript. Please find our final responses below in blue colour. We'd like to draw your attention to the most important changes:

1. As both reviewers pointed out, the former tile was not ideal and we have adapted it (see above).
2. We have added a more detailed explanation to the regional data set used in our study (L. 150 ff).
3. Figures 1 and 7 were adjusted to be more accessible.

All adaptations are indicated in the track-changes file.

Thank you for your efforts in handling this manuscript. We look forward to hearing from you soon.

Kind regards,

Jacob Hirschberg et al.

Response to referee comment by Ben Mirus

We thank Ben Mirus for providing a constructive review and his interest in our study. Here we respond to the comments and address how we will implement the changes in a revised manuscript.

**General comments**

*There are quite a few minor details that are missing in the abstract, most of which could be gleaned eventually from reading the entire paper, but should be included in the abstract for completeness.*

We have adapted the abstract according to the answers to the specific comments below.

*…I could not find information on how storms and ID are defined for the regional datasets, nor did I find a clear explanation that multiple durations for antecedent rainfall conditions were explored in the RF model.*

We have added these specifications according to the answers to the specific comments below.

*…some further details on the rationale behind the scenarios selected for different optimization strategies would be helpful.*

We have added details on the rationale behind the scenarios according to the answers to the specific comments below.

**Specific comments**

*I found the title quite critical, when actually the paper is not only about limitations, but rather a comprehensive evaluation of multiple approaches to debris-flow forecasting. Perhaps the title could be revised to more fairly represent the important contributions of this work.*

We agree that the title is a bit unrewarding. We have changed the title to "Evaluating methods for debris-flow prediction in an Alpine catchment".

*L3. I'm not sure I agree that there are no standardized procedures. I think it's more reasonable to state that there are multiple competing methods that have not been objectively and thoroughly compared at multiple scales.*

We have adapted the sentence according to the reviewer's suggestion.

*L6. Consider stating "record duration" since you are talking about time, not a distance (length).*

That makes sense. We have changed all expressions "record length" to "record duration".

*L12. Regional landslide dataset with local rainfall input or with a regional rainfall database? This is a critical detail that needs to be clarified in the abstract even if it can be determined later in the paper.*

The regional landslide database is combined with a regional rainfall database, i.e. with a gridded interpolated daily rainfall from stations. We have specified this.

*L13. If these implications are important, is it important enough to list them in the abstract? Also, state here whether the RF model was tested for just local or also for regional?*

We left the reader in the dark here. We added that an important finding is that the appropriate method depends on the available data sets.
The RF model was only tested for the local case. We added this information to the text.

*L15. I found this "30-min maximum accumulated rainfall" a bit confusing as it isn't really standard terminology. Is this the greatest accumulated depth of rainfall observed within a given 30-minute period of a storm? If so, wouldn't that be basically equivalent to the peak 30-minute rainfall intensity (I-30)?*

Yes, it is the peak 30-min rainfall intensity. We have changed the term here and in the rest of the manuscript.

*L17. Increase in predictive performance over which other threshold optimization approach/approaches?*

RF yields a slight increase over the ID-threshold. We adapted the sentence.

*L41. Again, it's not that there are none, but that a few established procedures are in use and that those approaches have not been compared objectively and thoroughly.*

We adapted the sentence accordingly.

*L82. Could also mention that you evaluate these differences for both a local vs. regional landslide inventory.*

Thanks, we added it.

*Figure 1. Legend should explain what the blue shaded channel and also the X marks the Illhorn peak. It wouldn't hurt to put the elevation of the Illhorn and the force plate or catchment outlet to provide easy reference for the steepness of the basin.*

We added these specifications.

*L143-146. Consider briefly explaining the gridded daily rainfall product, including the spatial resolution and how it is collected/calculated, as well as what rainfall value was used for the threshold evaluations (i.e., did you use rainfall values from the nearest grid cell, or some grid-cell averaging, or ...?). This is important context for evaluating the ID thresholds at the regional scale vs. local scale.*

The rainfall dataset for the regional analysis consists of a 1km x 1km gridded product of daily precipitation sums obtained by interpolation of rain gauges (ca. 420), accounting for local climatology and precipitation-topography relationships. We have added these specifications.

*L173-202. I guess you didn't explain the regional data here in Table 1. Perhaps that's not necessary, but you do need to define your MIT for the daily/regional data analysis. How are multi-day storms determined?*

The MIT for daily rainfall is 1 day and multi-day storms consist of a sequence of wet days. This definition of events overestimates event duration and underestimates mean intensity compared to hourly data, but compensates for this by longer and more dense records (see Leonarduzzi and Molnar, 2020). We added these specifications.

*L192. Initially, I assumed that this 3-90d antecedent conditions meant the cumulative rainfall total measured between 3 and 90 days prior to the storm event. While there needs to be some explanation of why 3 days was selected as a cutoff (why not 2 or 1 day?), there also needs to be a clearer explanation that multiple potential durations of antecedent rainfall were considered. This only becomes apparent in Figure 7 and the associated analysis of the RF results and variable importance.*

Your assumption is correct. We cut off at 3 days because it didn't make any difference in the results. However, we added 1- and 2-day antecedent rainfall for the sake of completeness. We specified that max rainfall intensities and antecedent rainfall was computed and tested for a range of values within the indicated boundaries.

*L208. Yes, and thus does not consider the rate of false alarms.*

We added this clarification.

*L204-216. These paragraphs could benefit from an explanation of the shape parameters in terms of how they influence ID threshold shape/position, and then subsequently the rational for why the two contrasting optimization approaches (LR-TSS vs, TSS-TSS) were selected. It might not be clear to all readers the significance of these choices.*

We added the ID equation so the parameters can be inspected directly. We also added an explanation saying that in the log-log space α controls the intercept and β the slope of the threshold line.

We also stressed that the fundamental difference in the two approaches lies in the way β is determined and that we therefore can expect different values and sensitivities of β for these methods depending on the data set.

*L223. Consider clarifying the "... original complete (or 17 year) record..."*

We added this clarification.

*L248. By "classical" ID thresholds, you mean those optimized with ROC statistics (LR-TSS, TSS-TSS)?*

*Yes, we clarified this.*

*Figure 3. Difficult to see what the minimum number of debris flows are in each month, but it looks like they're all zero. If so, consider just stating in the figure caption. (b) also, see previous comment about maximum 30-min accumulation. Isn't this just more or less equivalent to the peak I-30 (i.e. 7.2mm/h)?*

We improved the visibility of the number of debris flow. Yes, it is peak I-30 and we changed the term (see also answer above).

*L256-257. If seasonal snowmelt is a relevant control on rainfall triggering, then the antecedent precipitation variable ought to somehow account for this, but I suspect it cannot.*

We do not have direct evidence that snowmelt is a relevant control on debris-flow triggering (L. 128). Accounting for antecedent wetness, including infiltration from snowmelt, would require hydrological modelling since we don't have any measured data on snow or soil moisture. We hoped that the seasonal thresholds in Fig. 4b-c would shed some light on this issue, but the results were not conclusive, as we discuss in L. 300 ff.

*L260-261. Again, these non-conforming observations might also be related to the fairly coarse consideration of antecedent rainfall.*

We added shorter time scales of antecedent rainfall (1 and 2 days) to the analysis (see answer to L. 192). It did not affect the results.

*L266-268. As a discussion point, it could be interesting to compare this range in parameter variation to the ranges of typical ID thresholds reported in the literature, say for example the difference between values for Caine vs. Guzzetti et al. ID thresholds. I have not done this comparison myself, but it could be worth looking at.*

Bel et al. (2017, Fig. 12) did a thorough comparison of ID thresholds for debris-flows torrents. We referred to this study here.

*L372. Even though 20% seems low, this is actually pretty good performance overall for an ID threshold relative to others developed worldwide, so that just further highlights the multitude of complex interactions that lead to debris-flow triggering and justify the need to explore more data-rich approaches like the RF you propose.*

Thanks, we added this observation to the discussion.

**References**

Leonarduzzi, E. and Molnar, P.: Deriving rainfall thresholds for landsliding at the regional scale: Daily and hourly resolutions, normalisation, and antecedent rainfall, Natural Hazards and Earth System Sciences, 20, 2905–2919, https://doi.org/10.5194/nhess-20-2905-2020, 2020.

Response to referee comment by Clàudia Abancó

We thank Clàudia Abancó for providing a constructive review and for her interest in our study. Here we respond to the comments and address how we will implement the changes in a revised manuscript.

**Specific comments:**

*Title: It may be a bit misleading. It does not deal with the limitations of the thresholds but more with the uncertainties on their definition? I would suggest reconsidering it...*

We agree that the title is a bit unrewarding. We have changed the title to "Evaluating methods for debris-flow prediction in an Alpine catchment".

*L65: I would suggest adding a few references of studies using different MIT, as it is said they range from 10 min to 6 h but no references are given (although they appear later, in 3.3., but I would add them here too)*

We added references, including the ones mentioned in section 3.3.

*L82: Although the two methods that are going to be compared are mentioned in the abstract, I would list them here too*

We have added a sentence regarding the methods: "These methods use linear regression and/or True Skill Statistic to determine the ID-threshold parameters alpha and beta."

*L80-85: I miss here stating as an objective (maybe as a secondary one) the analysis of the performance with local vs. Regional dataset, which is stated in the abstract.*

Thank you for pointing this out. The local vs. regional analysis should be mentioned here. We have added that the differences of the two methods were evaluated both for a local and for a regional data set.

*L89-98: Cite Figure 1 in this paragraph*

We added the reference to figure 1.

*L106: Rain gaugeS in plural? If there's more than one, why in the Figure only 1 is shown? Why only data from 1 is used?*

There are two other gauges in the basin which are not suited for this study. One has been moved during the study period and the other is sheltered by trees. Furthermore, the used rain gauge is the one closest to the triggering area. We added this explanation.

*L111-113: If the geophones and depth sensors have been removed, which sensors (less maintenance) have been installed?*

We created a bit of confusion here. Badoux et al. (2009) describe the alarm system and some of these sensors have been replaced. However, these are not the same sensors used for the debris-flow monitoring conducted by the research institute WSL, which this paper focuses on. Here, only information on the time of debris-flow occurrence was used as it was detected at the force plate or earlier by geophones upstream. We stated more clearly that the alarm system and the observation system work independently, and indicated the sensors used in this study in Figure 1.

*L115: By including the citation of Badoux in line 113 I think you could delete it in Line 115, it is clear for the Reader that details can be found there*

We deleted this last sentence as it is redundant.

*L122: Any reference where snowmelt has been observed (even if not as sole trigger)?*

We added that although snowmelt in many places adds considerable amounts of liquid water to the debris (Mostbauer et al., 2018), it has never been observed to be the sole trigger in Illgraben.

*Figure 1: Why only force plate is indicated? I would suggest adding the other sensors (the new ones replacing the geophones)?*

We added the locations of the relevant sensors and label them (see also response to L111-113).

*L135: 5 mm? This sounds like a very low number...*

It does sound low but the Illgraben ID thresholds are also lower than in other places. Debris flows are either triggered in individual gullies or in the main channel. When the latter is the case, water drained from multiple gullies with low infiltration rates concentrates in the channel. This possibly explains the low thresholds.

*L136: Does the local rain gauge not have a thermometer? Also, I would suggest moving Temperature, lighning strikes and other parameters to another paragraph, as I understand these are all variables for the Machine learning, but not actually for the main ID thresholds comparison? I was a bit confused reading about rainfall and changing to temperature abruptly, as it's the first time you mention the temperature variable*

There is a thermometer at the local rain gauge but because the sensors are not properly shielded, the measurements are unreliable. We added this explanation.
We made two paragraphs to better separate variables only used for machine learning.

*L160: Reference of TSS?*

We added references for the alternative names of the score. We think that in the landslide community TSS is the most common name, as in the studies cited before the equations.

*L169: Include Area Under Curve as clarification of what AUC stands for*

For all used abbreviations, we capitalized the letters standing for the abbreviation.

*L173: I understand that you used the same criteria for both trigg and non trigg rainfalls?*

Triggering events end with landslide/debris-flow occurrence. We have specified this.

*L175: Delete ? before Deganutti*

LaTeX couldn't find the reference due to a typo. We corrected this.

*L180 and Figure 2: I think this is more results than methods?*

We decided to add it to the method part because it was a necessary step in the process of defining rainfall events and in the result section we wanted to focus on the results addressing our research questions. As mentioned in the text (L. 197), we followed Bel et al. (2017) for this purpose, who thoroughly discusses ID threshold sensitivity to MIT.

*Figure 2 (b): Sensitivity? This word may be confused by SE? Could it be called "Analysis of ID-threshold parameters with changing MIT"?*

Good point. We followed the reviewer's suggestion.

*L184: I would not say that β stabilizes, but reduced the increasing tren dat MIT 3h... (in Fig 2b)*

We adapted the sentence accordingly.

*L197: Actually, if it was snowing the data from the rain gauge would not be valid, right? As it is not heated... Have you considered this? If so, maybe you could mention here.*

Yes, this is an additional reason not to trust the rainfall measurements when it was cold. However, most debris flows occur in summer when solid precipitation is rare even for the higher parts of the basin. Therefore, there are only very few data points where this is the case. We added this statement.

*L208: This last sentence of the paragraph ("Lately, confusion matrix...") is actually a bit confusing to me. You have not used frequentist method, right? You used linear squares and LR&TSS and TSS&TSS methods if I have understood right. Therefore, the sentence is confusing as it seems that you have calculated the confusion matrix and ROC for the frequentist method...*

In this paragraph we reflect on how ID threshold parameters are determined in literature. We clarified that determining ID threshold parameters using confusion matrix and ROC, on which we rely on in this paper, is an alternative to the frequentist method.

*L220: This is also confusing. A record of length 5 years includes 5 annual samples that can include repetition of the same year? Why is the procedure repeated 100 time for each record length? Please clarify*

We agree that the sentence is a bit confusing. The procedure is resampling with replacing and is frequently used for uncertainty assessment. The replacing is fundamental to the uncertainty assessment. Without it, we wouldn't be able to estimate uncertainty bounds because for 17 years there would only be one possible combination. We have clarified this and added a reference to the resampling method.

*L280: I think it would be good to see the total rainfall amounts at some point in a Figure, as it is stated here that long duration need more rainfall (logic, but still nice to see)*

We added the total rainfall amounts to Figure 4b-d.

*L297: Higher antecedent rainfall amount may lead to higher degree of pore saturation along the entire channel bed, but also, in some cases the antecedent rainfall would mostly contribute to the generation of lateral flow and increase of water table (e.g.: M.N. Papa, V. Medina, F. Ciervo, A. Bateman 2013, Derivation of critical rainfall thresholds for shallow landslides as a tool for debris flow early warning Systems). This could also correlate with the fact that magnitudes are bigger, but I would say that the correlation between the antecedent rainfall and the magnitude it is a tricky point and needs careful evaluation...*

Thank you for pointing this out. We added it to the discussion. However, in Illgraben in Hirschberg et al. (2019) we found that 14 days antecedent rainfall best correlated with the magnitudes, and we also tested shorter durations. Our thinking is that in such a steep gully, the rainfall contributing to lateral flow certainly happens at shorter time scales which is closer to few hours rather than 14 days. This gives us confidence that antecedent rainfall influences the saturation in the channel bed, and therefore the debris-flow magnitudes. We also added this reasoning to the manuscript.

*L305: TSS&TSS thresholds are lower for short durations (<4.5 h) and higher for long durations- after this I would add (Figure 5e)*

We added it.

*L311: However, the biases decrease to _30% already after 6 years or _25 triggering events- For both? Or only for β? I can't see it that clearly in alpha?*

For both. We matched the y-axes in order to make it clearer.

*L335: Also, the source of rainfall data is different, right? If I am not wrong the work of Leonarduzzi et al. it was not based only in rain gauge data. Therefore, apart from climàtic, topographic and lihologic uncertainties it may be also from the type of rainfall data?*

Yes, you are right. We also added a better description of the regional dataset (as suggested by RC1). The rainfall dataset for the regional analysis consists of a 1km x 1km gridded

product of daily precipitation sums obtained by interpolation of rain gauges (ca. 420), accounting for local climatology and precipitation-topography relationships. Findings relevant to this study by Leonarduzzi and Molnar (2020) regarding daily vs. hourly rainfall are also mentioned e.g. in L. 60 or in L. 192.

*Figure 7:*
- *The grey dots are very difficult to see, specially over blue, red and green bars. Change colour of bars or make dots bigger*
- *I find this figure particularly dense and a bit difficult to follow. Some ideas on how it could be made a bit easier to read:*
    - a) *I understand that RF_ID+1 is based in one sigle predictor (the one with best performance). Why not indicating which one instead of leaving the reader to interpret?*
    - b) *Same with RF_ID+var and 4 predictors*
    - c) *Maybe then it would not be necessary to include all the single predictors in the same figure. Either include them in a separate figure or as supplementary material?*
    - d) *If you think it is relevant to keep the same format, I would suggest indicating the selected predictors for each RF model in some way...*

We followed the reviewer's suggestion and changed the colours and marker size to make the figure more accessible. it's relevant that all used predictors remain in the figure to directly see the best predictors without having to look for the irrelevant ones in the text. However, we changed the colours so that it's easier to distinguish between the important and unimportant predictors. We also adjusted the labels of some models to be more intuitive and to be more consistent with the literature.

**References**

Hirschberg, J., McArdell, B. W., Badoux, A., and Molnar, P.: Analysis of rainfall and runoff for debris flows at the Illgraben catchment, Switzerland, in: Debris-Flow Hazards Mitigation: Mechanics, Monitoring, Modeling, and Assessment - Proceedings of the 7th International Conference on Debris-Flow Hazards Mitigation, pp. 693–700, 2019.

Leonarduzzi, E. and Molnar, P.: Deriving rainfall thresholds for landsliding at the regional scale: Daily and hourly resolutions, normalisation, and antecedent rainfall, Natural Hazards and Earth System Sciences, 20, 2905–2919, https://doi.org/10.5194/nhess-20-2905-2020, 2020.